# End of Life in Italy: Ethical and Legal Perspectives

**DOI:** 10.3390/healthcare13060666

**Published:** 2025-03-18

**Authors:** Rosagemma Ciliberti, Linda Alfano, Chiara Robba, Nicolò Antonino Patroniti

**Affiliations:** 1Section History of Medicine and Bioethics, Health Science Department (DISSAL), University of Genoa, 16126 Genoa, Italy; rosagemma.ciliberti@unige.it (R.C.); alfanolinda65@gmail.com (L.A.); 2Dipartimento di Scienze Chirurgiche Diagnostiche ed Integrate, University of Genoa, 16126 Genoa, Italy; nicoloantonino.patroniti@unige.it; 3IRCCS Policlinico San Martino, 16132 Genova, Italy

**Keywords:** euthanasia, assisted suicide, advance treatment declarations, self-determination, Constitutional Court, life-sustaining treatments, bioethics

## Abstract

The regulation of end-of-life decisions has been the subject of intense debate for years, marked by the challenge of reconciling two fundamental ethical principles: preservation of life and individual self-determination. From a legal perspective, numerous court rulings have outlined an evolving framework, highlighting the difficulty of establishing a regulatory approach that balances constitutional rights with ethical values. This study examines key Italian judicial decisions, with a particular focus on recent Constitutional Court rulings regarding end-of-life issues, and discusses the underlying ethical and humanistic perspectives. We aim to explore the key ethical and legal issues arising in the context of end-of-life regulation. Judicial developments demonstrate an increasing recognition of individual self-determination in accessing assisted suicide despite persisting ongoing ambiguities and regulatory gaps. The end-of-life debate underscores the urgency of moving beyond abstract and schematic approaches, favoring a perspective that integrates multidisciplinary expertise and human sensitivity. Ensuring effective access to palliative care and comprehensive social and healthcare systems is essential to alleviate suffering and provide genuine alternatives to assisted suicide.

## 1. Introduction

Over the last decades, end-of-life issues have gained increasing prominence in the Italian public discourse, highlighting the complex interplay between the principle of preserving life and that of individual self-determination. This has raised profound ethical questions about how to balance fundamental but often conflicting values [1]. It is crucial to recognize that end-of-life decisions encompass distinct issues, such as euthanasia, assisted suicide (AS), and advance directives (ADs), each one with unique legal and bioethical implications.

This debate unfolds within a context marked by ethical ambiguities arising from the technological extension of survival, often in conflict with individual value systems, and by a cultural tendency to deny death, obscuring the significance of human finitude and fragility [2,3,4].

Euthanasia refers to the act, carried out by a physician (or another individual), of intentionally ending a person’s life at their explicit and voluntary request to alleviate suffering from a terminal illness with a poor prognosis. It can be classified into two types: active euthanasia, which occurs when the physician (or another individual) directly induces death, typically by administering a lethal injection, and passive euthanasia, in which death occurs indirectly by withholding or withdrawing life-sustaining treatment.

However, certain cases are not considered euthanasia. Specifically, deaths resulting from the refusal or withdrawal of life-sustaining treatment by a competent and conscious patient do not fall under euthanasia. Similarly, premature deaths that occur in accordance with the bioethical principle of double effect—wherein an adverse outcome, such as death, is tolerated if it arises as an unintended consequence of a treatment aimed at relieving suffering, such as appropriate palliative care—are also not classified as euthanasia.

In assisted suicide, a physician intentionally aids another person in ending their own life or provides the means to do so, strictly at the individual’s explicit and voluntary request, after the person has freely and autonomously decided to pursue suicide (for example, death resulting from ingesting the pills prescribed by the physician is assisted suicide).

In light of these ethical and legal dilemmas, the legal status of euthanasia has become a central topic of global debate.

The legal status of euthanasia varies significantly across the world, reflecting profound cultural, religious, and ethical differences. In countries such as the Netherlands, Belgium, Luxembourg, Canada, and Spain, euthanasia is legal under specific conditions, typically requiring the patient to be experiencing unbearable suffering with no prospect of improvement [2]. Euthanasia is also legal in Colombia, Australia (in the states of Victoria, Western Australia, and Tasmania), and New Zealand [3]. However, it remains illegal in many other countries. The number of countries where euthanasia and assisted suicide are legal, however, is steadily increasing, and in these regions, the number of individuals choosing these options is also on the rise [4].

Some jurisdictions only allow assisted suicide under strict regulations. Switzerland, for instance, permits assisted suicide but not euthanasia, provided that the act is not motivated by selfish interests. Other countries, such as Germany and Austria, have recently revised their legal frameworks on assisted dying, reflecting evolving societal perspectives on the matter. Colombo and Dalla Zuanna [4] observe that in several countries where euthanasia and assisted suicide have been legal for years, their incidence as a proportion of total deaths continues to vary significantly.

In Italy, euthanasia remains illegal, and assisting in suicide is punishable under Article 580 of the Italian Penal Code. However, recent legal developments have introduced some degree of flexibility into the legal landscape.

Numerous rulings have drawn significant attention, both in the judiciary and the media, to dramatic cases such as those of Eluana Englaro, Piergiorgio Welby, Fabiano Antoniani (known by his stage name DJ Fabo), Fabio Trentini, and Federico Carboni (known by the pseudonym “Mario”) [1,2]. These cases, while varied in their specifics, share the common thread of confronting the profound human, social, and legal dilemmas underlying the tragedy of death. End-of-life issues transcend the boundaries of medicine and law, presenting a multidimensional challenge that intersects with fields such as anthropology, philosophy, and religion. They provoke deep reflections on the meaning of life, individual autonomy, and the value of suffering [3].

In 2019, the Italian Constitutional Court ruled on the case of DJ Fabo, acknowledging that, under certain circumstances, assisting suicide might not be punishable. The ruling established that individuals with an irreversible medical condition who experience unbearable suffering and are fully capable of making autonomous decisions may request assisted suicide, provided the procedure takes place within a public healthcare framework and is under ethical scrutiny. This decision marked a significant turning point, yet it did not lead to a comprehensive legislative framework regulating euthanasia or assisted suicide. As a result, Italy remains in a legal gray area where judicial interpretations play a crucial role in defining the boundaries of end-of-life treatment.

### Decision-Making

The debate surrounding the legalization of euthanasia and assisted suicide across different countries takes place within the broader context of evolving public attitudes toward end-of-life decisions.

A recent demographic study conducted on 13 countries (8 of them in Europe) in which these practices are legalized found that the majority of the population expressed clear support for assisted suicide and euthanasia as an option to put an end to ‘intolerable pain’ [5] (Table 1, Figure 1).

This shift reflects long-term cultural transformations, particularly in developed nations, where perspectives on suicide have changed significantly over time. However, in countries with stricter legislation, prevailing cultural norms tend to be less permissive toward these practices. Empirical research does not indicate a uniform trend but highlights significant variations both across countries and among individuals [4]. Acceptance of euthanasia and assisted suicide is generally higher among those with greater socioeconomic and educational status, whereas it tends to be lower among individuals for whom religion plays a central role. Interestingly, experiencing severe physical suffering does not necessarily correlate with greater support for these practices, while individuals facing economic and social vulnerability are often more likely to oppose them.

From a cultural perspective, attitudes toward end-of-life decisions are shaped by the broader societal context. Countries with well-functioning healthcare systems and strong democratic institutions generally demonstrate greater acceptance of euthanasia and assisted suicide. Conversely, in societies where healthcare is unreliable, and governance is more authoritarian, opposition to these practices is significantly stronger frameworks. This underscores that end-of-life decisions are not merely ethical or personal choices but are deeply embedded in cultural, social, and institutional.

In this context, it is crucial to recognize that end-of-life decisions encompass distinct issues, such as euthanasia, assisted suicide (AS), and advance directives (ADs), each one with unique legal and bioethical implications.

These terms, often ambiguously used outside the scientific community, require precise definitions to foster informed and meaningful dialogue. Only through a clear understanding of the underlying issues can society actively engage in discussions that profoundly impact ethical and social dimensions, reinforcing a collective sense of responsibility on matters of shared significance.

This debate unfolds within a context marked by ethical ambiguities arising from the technological extension of survival, often in conflict with individual value systems, and by a cultural tendency to deny death, obscuring the significance of human finitude and fragility [1,2,3].

However, the issue of euthanasia and assisted suicide (ASE) remains a highly debated topic, involving various viewpoints rooted in ethical values. Proponents of these practices defend them as compassionate approaches to alleviating extreme suffering, highlighting principles like beneficence and autonomy.

On the other hand, critics argue that both euthanasia and assisted suicide contradict medical ethical codes and the Hippocratic Oath, harm the doctor–patient relationship, erode public confidence in healthcare systems, and violate the principle of non-maleficence. A significant concern raised by these opponents is the “slippery slope” argument, which suggests that legalizing euthanasia and/or assisted suicide could lead to an uncontrollable spread of these practices, resulting in mistakes, exploitation, and violations of the rights of the most vulnerable individuals.

Religious opposition also plays a significant role in this debate, with many citing the sacredness of life as a key principle. Critics also refer to the concept of justice, emphasizing the need for fair access to care and the safeguarding of individual autonomy. Furthermore, they highlight the psychological and ethical burden placed on healthcare providers who may be asked to end a patient’s life. Ultimately, decisions regarding end-of-life care involve not only legal considerations but also fundamental ethical questions, requiring thoughtful deliberation to protect both patient rights and the integrity of the healthcare system. These ethical considerations should guide the practice of euthanasia and assisted suicide, as outlined in Table 2.

## 2. Methodology

This analysis is based on a scoping review of key legal rulings, official legislative documents, and ethical guidelines relevant to end-of-life decisions. Court judgments were selected based on their legal significance, particularly those issued by the Constitutional Court and other high judicial bodies addressing euthanasia, assisted suicide, and life-sustaining treatments. Official documents, including national laws and parliamentary reports, were examined to provide a comprehensive legal framework. Ethical considerations were integrated through the analysis of guidelines issued by medical and bioethics committees, as well as scholarly discussions on patient autonomy and dignity. This approach ensures a balanced evaluation of both legal and ethical perspectives, offering a nuanced understanding of the complexities surrounding end-of-life care.

### 2.1. Advance Directives

The issue of individual choice in healthcare treatment received significant regulatory recognition in Italy with the approval of law No. 219 of 22 December 2017, entitled “Rules on informed consent and advance directives” [1,2].

Although this law did not represent a revolutionary shift on the issue of consent, it marked a turning point in the systematic organization of a subject that, supported by a solid and coherent legal framework (both at the national level, with constitutional principles, and at the international level, with human rights conventions and declarations), needed an update and a more significant recognition of every individual’s right to make healthcare choices.

Law No. 219 of 2017 also had the merit of providing legal certainty to a right that, although existing, had previously been difficult for citizens to enforce in practice. In fact, this law explicitly established that individuals have the right to refuse or discontinue any healthcare treatment, even if such a decision leads to extreme consequences, such as death, and that the physician respecting this choice is exempt from liability.

This principle is, in fact, already grounded in the Italian Constitution, particularly in Articles 2, 13, and 32, which guarantee the right to self-determination, including the ability not only to accept or refuse treatment but also to discontinue ongoing healthcare. However, until the early 2000s, as exemplified by the case of Piergiorgio Welby, these constitutional provisions struggled to find full practical application [1,2,3].

From a medico-legal and legal perspective, interrupting life-saving treatment exposed physicians to the risk of prosecution for murder (voluntary manslaughter or, at the very least, manslaughter of a consenting person). Psychologically, complying with a patient’s request to suspend treatment often heightened the physician’s sense of responsibility, making them feel more culpable for the harm or consequences, including death, resulting from that decision. This was further compounded by a conceptual difficulty in recognizing the ethical equivalence between withdrawing and withholding treatments, both of which reflect a patient’s wishes. Regarding ADs, the aforementioned No. 219/2017 represented a significant innovation in the Italian legal system, filling a legislative gap and ensuring effective protection of the patient’s right to express their wishes in advance concerning healthcare treatments, even in anticipation of a potential future inability to self-determine.

Ethically, AD legislation not only recognizes and expands the right to self-determination but also demonstrates a concrete focus on the principle of vulnerability, as enshrined in the Barcelona Declaration [1]. The ADs are, in fact, an essential protective tool for individuals in fragile conditions—whether temporary or permanent—who are unable to fully exercise their decision-making sovereignty regarding matters that concern them. They help preserve dignity and respect for the individual’s wishes while also acting as a safeguard against overly invasive or disproportionate medical interventions that may not align with the patient’s actual needs and preferences.

Furthermore, ADs hold both relational and preventive value, fostering dialogue among patients, physicians, and family members while offering a foundation for mutual understanding and clarity. They also serve as a protective tool, enabling healthcare professionals to demonstrate adherence to the patient’s expressed wishes and to avoid potential civil or criminal liabilities arising from the administration of unwanted treatments. In this context, ADs strengthen patients’ rights by offering healthcare providers clear guidance on the individual’s preferences.

However, critical issues arise when the decision-maker is not a person suffering from illness but rather a healthy individual without direct experience of the condition. In such circumstances, choices based on advance directives risk becoming abstract and lacking concrete substance. This limitation reduces the potential of advance directives to merely be tools for refusing or defending against invasive medical treatments rather than serving as instruments of regulation and means to fully express self-determination [1,2,3,4,5].

### 2.2. Prerequisites of the Therapeutic Choices

When addressing therapeutic choices, it is essential to reflect on the fundamental principles that ensure each decision is authentic, responsible, informed, and respectful of the person’s deepest and most intimate values. This reflection is crucial in all areas of healthcare but becomes especially important in the context of end-of-life decisions.

The primary reference here is the awareness of vulnerability, understood as the recognition of the intrinsic fragility of the human condition. This principle, vital for genuine self-determination, calls for an awareness not only of the vulnerability of others but also of one’s own, fostering a sense of relationship, solidarity, and shared responsibility [2]. Therefore, awareness of vulnerability necessitates an approach that values mutual support and empathy—key elements for guiding every decision-making process in a respectful and humane way [5].

In specific cases, this relational context can be concretized by facilitating the patient’s engagement with their loved ones and multidisciplinary care teams. This approach allows patients to feel heard, understood, and supported in making decisions about their care rather than simply following procedural guidelines. For example, in the context of advanced illness, conversations with family and healthcare providers about the patient’s wishes can enable a more personalized, compassionate decision-making process that respects both autonomy and relational context.

The second reference concerns the deeper meaning of care, perceived in the dual dimensions of “to cure” and “to care”. Alongside the technical aspect of treating illness, there is the aspect of “caring” for the individual in their complexity, considering their subjectivity, needs, desires, and existential dimension. This holistic approach requires a systemic and integrated view of the person, one that respects their identity, history, and the innermost aspects of those seeking assistance.

Curing without caring is, in fact, a contradiction in terms. Authentic care is not limited to advanced technology but is realized through proximity, listening, support, and human connection. This approach overcomes the persistent dichotomy between technical and human aspects, which should not be exclusively delegated to specific roles [1]. Finally, a central element in understanding the premises of choice lies in the meaning and purpose of contemporary medicine. Medicine is, and must remain, a humane profession that integrates biomedical science, scientific methods, and relational skills. Its primary goal is to understand the issues facing the sick person, provide effective treatments, manage pain, and offer comfort, accompanying the patient through the terminal stages of life and beyond. In this context, medicine cannot limit itself to treating the body or the disease but must consider the person in their entirety, respecting their dignity and choices. Only in this way can therapeutic choice be regarded as a truly free, conscious, and respectful process that takes into account the complexity of the human being.

In line with the premises outlined above, law 219/2017 stands out for the legislator’s intention to promote an integrated and comprehensive vision of care aimed at supporting the autonomy of the individual, especially during moments of vulnerability that may characterize their existence, whether temporarily or permanently. This concept of autonomy is not viewed as an inherent or taken-for-granted fact but rather as a goal to be achieved and strengthened through a gradual and relational process of support [2].

Many of the provisions in law 219/2017 reflect an approach that prioritizes supporting the patient’s autonomy, emphasizing integrated and relational care. These provisions include the role of the multidisciplinary team, psychological support for those refusing life-saving interventions, ADs, exemption from liability for physicians who respect the patient’s wishes, the importance of palliative care, pain management, continuous deep sedation, and time dedicated to communication. All these provisions converge on a common goal: to effectively support the individual in making an informed and authentic choice. Such a choice cannot be considered genuine if it is made without adequate information or in a context of abandonment and isolation, where procedural concerns overshadow the human dimension.

### 2.3. The Right to Choose the End of Life: Jurisprudential Evolution and Legislative Limits

The debate on the right to choose “to die” has gained unprecedented centrality in the Italian legal landscape since 2018, when the Constitutional Court, in ruling n. 207, urged Parliament to intervene with end-of-life legislation within a year. This ruling stemmed from the case of DJ Fabo and his assistance from Dr. Marco Cappato while traveling to Switzerland to access AS [1,2].

The legal issue centered on Article 580 of the Italian criminal code, a provision dating back to 1930, which imposes penalties ranging from 5 to 12 years for both incitement to suicide—understood as actions that encourage or reinforce a person’s suicidal intent—and material assistance to suicide, which involves providing the means or facilitating the execution of the act. In this context, the Court of Milan decided not to proceed immediately against Dr. Cappato but instead referred the matter to the Constitutional Court, asking it to assess the compatibility of this article with constitutional principles and pointing out the law’s inadequacy in addressing current needs and contexts [3].

The Constitutional Court recognized the partial inapplicability of Article 580 of the criminal code, deeming it unsuitable for regulating situations like that of DJ Fabo, where suicidal intent was independently and freely formed. However, the Court, mindful of the impact of its remarks and the values at stake, chose not to immediately declare Article 580 unconstitutional. Instead, it postponed the official discussion of the constitutional issue to the hearing on 24 September 2019, adopting a “collaborative” and “dialogical” approach, and entrusted Parliament with the responsibility of addressing the legislative gap through a specific law.

Despite the one-year deadline set by the Court, Parliament failed to adopt any legislative measures. As a result, the Constitutional Court intervened with Judgment n. 242 of 2019, a “manipulative” decision that partially altered the meaning of Article 580 [4,5].

The judgment declared the provision unconstitutional insofar as it failed to exempt from punishment those who assist a person in autonomously and knowingly deciding to end their life. This applies to individuals who are fully competent, reliant on life-support treatments, and suffering from an incurable condition causing intolerable distress. The conditions and procedures must be verified by a public health facility in consultation with the competent ethics committee.

This decision took on a surrogate role, compensating for the lack of a law that Parliament had failed to enact, despite the submission of five bills since then, none of which has been passed.

### 2.4. Contradictions and New Developments on the End of Life: The Case of Assisted Suicide in Italy

In recent years, the legislative and judicial process concerning end-of-life issues in Italy has been marked by legislative inertia and procedural chaos, with very few judicial decisions up until 2023. These rulings, often the result of lengthy proceedings in courts and appellate courts, have frequently compelled local health authorities to address patients’ requests to verify the necessary requirements for accessing AS [6].

On 24 July 2024, the Constitutional Court revisited the issue of AS in ruling n. 135, concerning the petition of a 44-year-old patient with multiple sclerosis seeking access to AS. The patient was accompanied by Marco Cappato and other activists to Switzerland. Notably, the patient was not reliant on life-sustaining treatments (LST) that would artificially preserve life but rather on continuous care that did not qualify as LST.

This case, which has also garnered the attention of the European Court of Human Rights, presents significant and complex challenges.

In Italy, the Preliminary Investigations Judge of the Florence Court has raised a constitutional legitimacy question regarding Article 580 of the Italian criminal code, as amended by Constitutional Court ruling n. 242/2019 [7]. The concern focuses on the provision that links the non-punishability of those facilitating AS to the presence of LST. This specific requirement in Italian law, absent in other jurisdictions regulating AS, has been challenged primarily for creating unequal treatment between clinically comparable conditions in terms of severity and suffering. The ruling emphasizes that the necessity for LST may be contingent on factors such as the nature of the disease, available therapies, or the patient’s personal choices (e.g., initial refusal of treatment). It is argued that conditioning access to AS on the presence of LST could result in inequality in the right to be free from suffering [3,5,8].

The judge of the Florence Court has argued that conditioning AS on the presence of LST would violate Articles 2, 13, and 32 of the Italian Constitution by unjustifiably restricting individual freedom. This restriction would compel patients to accept unwanted therapies in order to access SA. Furthermore, the requirement could force patients to unnecessarily prolong their suffering while awaiting clinical deterioration, which may be incompatible with their sense of dignity. According to the referring judge, dignity is a subjective parameter closely linked to quality of life, which the individual identifies through the exercise of their decision-making autonomy. Therefore, a patient suffering from an incurable and irreversible disease should have the right to end their life through SA if they perceive their existence as lacking value and meaning [4,9,10].

Additionally, the judge of the Florence Court raised the possibility of a conflict with the European Court of Human Rights, suggesting that the interference with the right to private and family life in this context is neither justified nor proportionate in relation to the protection of the right to life.

### 2.5. The Fragility of the Definition of Life Support

In the context of the Constitutional Court ruling, the definition of LST remains one of the most contentious issues in the bioethical and legal debate, marked by significant terminological and conceptual confusion over time. To date, there is not a clear, universally accepted definition of what constitutes LST. Neither Order No 207/2018 nor Judgment n. 242/2019 of the Constitutional Court has provided systematic criteria. Certain treatments, such as mechanical ventilation and dialysis, have gained consensus, while others—like artificial nutrition—have been the subject of years of legal disputes, culminating in the Englaro case (Cass. civ., sec. I, Judgment n. 21478/2007), and were later incorporated into law 219/2018 as medical treatments.

Regarding the criteria for identifying and distinguishing LST from ordinary treatments, it is important to note that, on 20 June 2024, the National Bioethics Committee (NBC) adopted a restrictive opinion on the criteria for defining LST. This opinion was issued shortly after the European Court of Human Rights ruled that the prohibition of assisted suicide in a country was not unlawful.

The majority of NBC members (19 in total) argued that LST should be understood as a treatment that directly replaces or substitutes vital functions, such that its discontinuation leads to death within a short time. According to this view, the LST criterion serves an essential bioethical function: preventing vulnerable individuals from experiencing undue pressure to end their lives, thereby reducing the risk of a slippery slope toward broader acceptance of assisted suicide. From this perspective, artificial nutrition and hydration would not be classified as LST because its withdrawal results in death only after a prolonged period rather than immediately or within a short timeframe. However, a minority of NBC members (5 in total) challenged this restrictive definition, arguing that life-sustaining support can also be provided through complex care plans that do not necessarily involve direct replacement of a vital function. This perspective emphasizes the patient’s subjective experience and the impact of ongoing treatment rather than an exclusively physiological definition of LST [11,12,13,14].

Additionally, a dissenting group of seven NBC members offered an alternative definition, asserting that LST encompasses any medical intervention—including pharmacological treatments or assistive measures—whose withdrawal, even if not immediately fatal, inevitably results in the patient’s death. According to this dissenting view, restricting access to assisted suicide based on the presence of LST creates an unjust and unconstitutional distinction (under Article 3 of the Constitution) between those whose lives depend on artificial support and those who endure severe suffering but are not dependent on such treatments [15,16,17].

### 2.6. The Decision of the Constitutional Court: Ruling n. 135/2024

In Judgment n.135 of 24 July 2024, the Constitutional Court addresses with both ethical and legal sensitivity the delicate issue of patient autonomy in the context of serious illness. It acknowledges the profound challenges faced by individuals suffering from advanced neurodegenerative diseases, which lead to near-total immobility and dependence on others for basic needs. The Court underscores how these conditions raise not only significant legal questions but also deep moral dilemmas, highlighting the tension between respect for life and respect for personal dignity.

Legal analysis in this judgment reveals the intersection between law and ethics, as the Court’s decision is not only based on legal principles but also considers the human implications of the decision to end one’s life. The balance between these two dimensions is necessary to ensure that the autonomy of the patient is respected within the confines of the law, which in turn should always reflect the ethical value of dignity and self-determination.

The decision emphasizes the need to balance the right to life with the freedom of self-determination, stressing that the patient must be in a state of full capacity, free from psychiatric disorders, and experiencing intolerable suffering that cannot be alleviated by palliative care (Table 1, Figure 1). The Court acknowledges that such conditions are frequently present in advanced stages of neurodegenerative diseases, thereby affirming a patient’s right to choose their own end-of-life path. According to the Court, the right to refuse medical treatment is inherently linked to the protection of bodily autonomy, serving as a safeguard against unauthorized external interference. This right, described as “negative liberty”, encompasses the refusal of complex, LST such as mechanical ventilation, artificial nutrition, and hydration. The Court asserts that there is no distinction between those already receiving LST and those who, although needing them, choose not to begin them: in both cases, the patient has the constitutional right to decide to end their life. The Court further emphasizes that it would be unreasonable to require a patient to accept life-sustaining treatments only to discontinue them shortly after merely trying to access AS. In both instances, the Constitution and law No. 219/2017 safeguard the right to end one’s life.

This decision further expands the definition of LST, specifying that it includes procedures such as bronchial suctioning, catheterization, and bowel management. While these procedures are typically carried out by healthcare professionals and require specialized skills obtained through formal training, they can also be learned by family members or caregivers who assume responsibility for the patient’s care [18,19,20]. Such procedures, essential for maintaining vital functions, fall within the category of treatments that individuals have the fundamental right to refuse, with the consequence of death occurring within a short time frame. Consequently, the Court affirms that these procedures must be considered LST for the purposes of applying the principles established in Judgment n. 242 of 2019.

In this regard, the Court makes a controversial equivalence, asserting that “all these procedures—just like artificial hydration, nutrition, or ventilation—can legitimately be refused by the patient, who thereby has the right to expose themselves to an imminent risk of death as a result of such refusal”. However, this equivalence fails to account for the fact that the cessation of assisted nutrition does not lead to death within a short time frame.

## 3. Ethical Considerations

The Supreme Court’s decision prompts a critical reflection on the tendency to address complex issues by imposing rigid frameworks. The requirement for strict criteria, such as the definition of life-sustaining treatments (LSTs), risks not only fostering unproductive debates in areas subject to rapid technological evolution but also falling short of adequately protecting vulnerable individuals.

Rather than relying on abstract medical classifications, it is crucial to implement tangible measures that ensure patients’ ability to make free and informed choices. In this regard, guaranteeing effective access to palliative care is of paramount importance, both for alleviating pain and for providing meaningful alternatives to assisted suicide (SA). However, significant disparities and deficiencies persist in the provision of palliative care and pain management services across Italy, with notable inequalities between northern and southern regions. These challenges are even more pronounced in the case of pediatric patients, underscoring the urgent need for equitable and comprehensive care.

Only one in three individuals who would benefit from palliative care actually receive it, with this percentage dropping to 15% among children. In Italy, two million people suffer from moderate-to-severe chronic pain, yet only 34% of them received at least one prescription for pain medication in the past year [7,20,21]. A stark illustration of these deficiencies is the failure to adhere to the obligation set forth in Article 11 of law No. 38/2010 (Provisions to ensure access to palliative care and pain therapy), which requires the Minister of Health to submit an annual report to Parliament on the law’s implementation. However, the most recent report sent by the Ministry of Health to Parliament dates back to 2019, underscoring the continued neglect of this critical issue.

The 2023 Budget Law established a target for palliative care to reach 90% of those in need by 2028. However, as of December 2021, two regions had not yet established an adult palliative care network, and eight regions had not yet set up a pediatric palliative care network.

In this context, it is crucial to analyze the ethical implications of limited access to palliative care (Figure 1, Table 1).

Studies indicate that in countries with well-developed palliative care systems, such as the Netherlands and Canada, requests for assisted dying are more frequently driven by existential suffering—such as loss of autonomy, dignity, or purpose—rather than uncontrolled physical pain. This finding is further supported by a study among Swiss professionals, which highlights the ethical and practical challenges associated with existential suffering as a motivation for assisted suicide and the varying degrees of acceptance of this rationale.

These findings underscore the urgent need to expand palliative care services to prevent scenarios in which patients seek assisted death simply due to inadequate symptom control. Strengthening palliative care infrastructure ensures that no patient is left feeling that death is their only relief. Moreover, persistent resistance to exceptional euthanasia among palliative care professionals suggests that rather than expanding medically assisted dying, efforts should focus on optimizing existing palliative care frameworks.

This strong resistance within the palliative care sector suggests that rather than legalizing exceptional euthanasia, healthcare policies should prioritize the full implementation of palliative care programs. By ensuring equitable access to pain relief, psychological support, and spiritual care, we can address the core drivers of assisted dying requests—particularly existential suffering—without prematurely resorting to irreversible interventions.

Only through a systematic reform of palliative care and a strong institutional commitment can true autonomy in end-of-life decisions be guaranteed [1].

The involvement of oversight bodies, such as medical commissions and ethics committees, plays a critical role in end-of-life care. Their purpose is to evaluate each decision on an individual basis, taking into account the human and personal context of each case. This is particularly important in a challenging reality characterized by a twofold issue: on the one hand, a systemic inability to manage urgent and extreme situations for those seeking assistance in making difficult choices; on the other, a tendency toward prejudicial resistance or even outright ostracism by certain institutions.

A debate, often abstract and theoretical, focused on defining what qualifies as LST risks diverting attention from the pressing need to build an operational support network. In particular, there is a pressing need to enhance the composition and expertise of Territorial Ethics Committees (TECs). Currently, these committees are primarily responsible for overseeing pharmacological trials, but they should be equipped with the technical and ethical qualifications necessary to handle the delicate and complex evaluations involved in end-of-life decisions [2]. Identifying broad, flexible, and coordinated operational criteria to respond promptly to requests for assistance represents an absolute priority.

Given the complexity of end-of-life issues, universal or simplistic solutions are not feasible. What is needed is an integrated approach that considers multiple perspectives and values, as well as empathetic listening and understanding for those facing such decisions. This requires moving beyond a purely technical or legal framework to include the existential and human context of each patient, as well as the emotional and moral burdens borne by family members, caregivers, and professionals involved [21,22,23].

Interdisciplinarity thus becomes a central element: only a dialogue among medical, ethical, psychological, and social experts can ensure that each decision is appropriately tailored to the specifics of the case. This approach must be accompanied by a profound awareness of the fragility inherent in such situations and a steadfast commitment to upholding the dignity and autonomy of the individual [24,25,26,27]. At the same time, it is essential for the institutions and bodies involved to foster an atmosphere of trust where different moral and cultural sensitivities are recognized within a climate of collaboration, dialogue, and respect.

## 4. Conclusions

Contemporary society presents complex ethical challenges that go beyond the mere application of rules. Human dignity, especially at the end of life, is not solely defined by biological survival but by the ability to live and die in alignment with one’s values and vision of life.

Therefore, a collective commitment is required to build a healthcare and cultural system capable of recognizing the complexity of the individual and providing tools to manage suffering without imposing standardized or authoritarian choices.

It is crucial to enhance access to palliative care and establish a comprehensive socio-healthcare support system, both indispensable for alleviating suffering and providing meaningful responses to critically ill patients. The involvement of bodies such as ethics committees equipped with the necessary expertise should be considered a fundamental step toward ensuring evaluations that respect the complexity of individual situations.

A balanced dialogue between technical rigor and human sensitivity must be fostered, one that upholds the values of dignity and self-determination in order to address these delicate and definitive issues appropriately. Such an approach should not only regulate but also accompany those faced with profound and often dramatic decisions, offering genuine support that respects diverse moral and cultural sensibilities.

## Figures and Tables

**Figure 1 healthcare-13-00666-f001:**
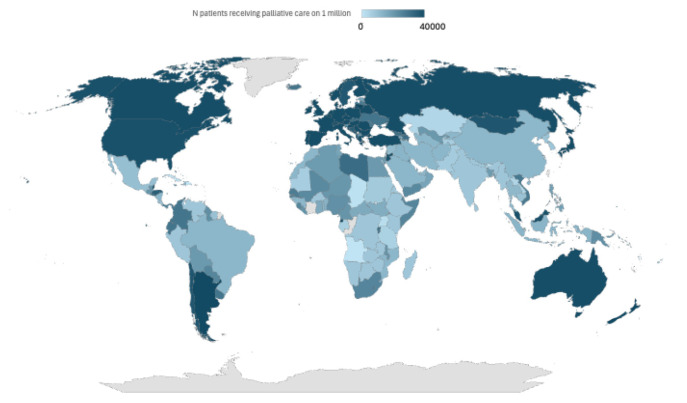
Heterogeneity across the world about the availability of palliative care.

**Table 1 healthcare-13-00666-t001:** Percentages of patients receiving palliative care or access to opioids for pain management, according to the World Health Organization (WHO).

Income Group According to WHO	Patients Receiving Palliative Care	General Availability of Opioids (% of Member States Reporting General Availability in Public Primary Care Facilities)
High income	69%	86%
Upper-middle income	27%	40%
Lower-middle income	3%	17%
Low income	1%	13%

**Table 2 healthcare-13-00666-t002:** Legal provisions and ethical principles.

Legal Provisions	Ethical Principles	Alignment with Clinical Practice
Patient consent laws.	Autonomy.	Respecting the patient’s right to make decisions regarding their own healthcare.
Informed consent is required before any medical procedure.	Autonomy refers to the right of individuals to make their own decisions.	Ensures that patients are informed about the risks, benefits, and alternatives of treatment.
Confidentiality and data protection laws.	Confidentiality.	Protects patient privacy and ensures their personal information is not disclosed without consent.
Laws governing medical confidentiality (e.g., HIPAA in the U.S.).	Confidentiality requires keeping personal health information private.	Clinicians must safeguard patient information unless consent is provided for disclosure.
End-of-life decisions.	Beneficence.	Healthcare providers must act in the best interest of the patient, especially in end-of-life care.
Legal guidelines for advance directives, euthanasia, and assisted suicide vary by jurisdiction.	Beneficence entails acting in ways that benefit the patient and promote their well-being.	Clinicians should make decisions that prioritize the patient’s well-being, particularly in critical care.
Non-discrimination laws.	Justice.	Ensures equitable treatment for all patients, regardless of background or status.
Laws that prohibit discrimination based on race, gender, disability, or socioeconomic status.	Justice in healthcare refers to fair and equitable distribution of resources and treatment.	Healthcare providers should offer equal care and access to treatment for all patients.
Mandatory reporting laws (e.g., abuse, neglect).	Beneficence and justice.	Ethical duty to report harm while balancing patient autonomy and societal protection.
Legal requirements for reporting suspected abuse or neglect.	Balances the duty to protect vulnerable individuals with respect for their autonomy.	Clinicians must report to authorities when necessary to protect vulnerable patients from harm.
Healthcare accessibility laws.	Non-maleficence.	The principle of “do no harm” is reflected in ensuring that healthcare is accessible and safe for all.
Laws ensuring healthcare services are available to everyone, preventing harm caused by lack of access.	Non-maleficence dictates the obligation to avoid causing harm.	Providers should ensure that their practices do not harm patients by either action or inaction.
Protection of vulnerable groups.	Vulnerability (Barcelona Declaration).	Provides additional safeguards for individuals who are especially vulnerable.

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
