# Peer review of "End of Life in Italy: Ethical and Legal Perspectives"

_healthcare, 2025, doi:10.3390/healthcare13060666_

Round 1

Reviewer 1 Report

Comments and Suggestions for Authors

Thank you for the opportunity to review this manuscript. The manuscript “End of life in Italy: ethical and legal perspectives” presents a well conducted study that explore the key ethical and legal issues arising in the context of end-of-life regulation. Considering the broader context of presented studies and further impact of the published papers, I have few suggestions suggested to be addressed before next steps in the publication.

  1. In the introduction section, I would suggest to include the current status of Euthanasia worldwide and particularly in Italy.
  2. Also, it would be good to include cultural perspective about end-of-life care. Some statistics about studies conducted about general public views concerning Euthanasia.
  3. Similarly, before moving into details of legal perspectives it would be good to include details of Euthanasia particularly its different forms i.e., active, passive etc.
  4. I would also suggest some perspectives in favor and against Euthanasia.
  5. Also, these examples/points can be well illustrated with the help of general examples from the current literature.

I consider this manuscript to be of interest; however, before publication, I would ask the authors to draw attention to the above mentioned points which, in my opinion, could be improved to improve the overall perception of the manuscript.

Minor comments:

I would suggest replacing/removing “End of life” in key words. As it has already been used in the title of the manuscript.

Author Response

Thank you to the Reviewer.Please see the pdf attached

Reviewer 2 Report

Comments and Suggestions for Authors

Dear authors, I read with great interest your contribution. See hereby my remarks:

  • the contribution balances between legal and ethical considerations, and they are not really convergent. Suddenly, you step over to the "ethical" (I completely agree with your statements), but where is the link with the very long legal analysis before?
  • I agree that we need a clear definition of the terminology: euthanasia, assisted suicide, advance directives...but I do not find in your contribution a discussion about euthanasia. You speak about assisted suicide, but where is the difference with euthanasia?
  • I admire your pleadoyer for a relational context (wherein autonomy can be taken up), but how do you make this concrete in specific cases?
  • You give an excellent analysis of care ((124-130)
  • The differentiation of life sustaining treatments is extremely difficult....I think you can bring more clarity than you do now.
  • the 179-188 sentence is really too long to be understandable and readable.    

Author Response

Thank you to the Reviewer, please see attached the response to your queries attached in the pdf file

Reviewer 3 Report

Comments and Suggestions for Authors

Dear Authors, I appreciate this manuscript for providing a thorough, detailed overview of the ethical and legal perspectives on end-of-life care in Italy. The discussion of jurisprudential developments and the focus on palliative care are particularly commendable. Nevertheless, several points could further strengthen this work:

1. Include Broader Global Statistics

Given that end-of-life issues concern healthcare systems worldwide, the manuscript would benefit from more detailed statistics on end-of-life care—both globally and in Italy. Providing specific data (e.g., percentages of patients receiving palliative care, access to opioids for pain management, regional disparities) would illustrate the scope of the problem and help readers appreciate the magnitude of unmet needs. A concise table summarizing these statistics might be especially useful.

2. Discuss the Ethical Implications of Limited Palliative Care

An important consideration is whether countries lacking adequate palliative care, including access to opioids and other pain-relief measures, have an ethical footing for legalizing euthanasia or assisted suicide. I guess unbearable pain is a most common reason patients may seek to end their lives prematurely. Expanding on how effective pain control and robust palliative support could potentially reduce requests for assisted death would add depth to the discussion.

3. Strengthen Methodological Transparency

The manuscript predominantly focuses on a legal-ethical analysis. However, it would benefit from a concise description of the authors’ methodology—specifically how court rulings, official documents, and ethical guidelines were selected and analyzed. Even a brief methods section would enhance transparency and academic rigor.

4. Provide a Summary Table of Legal Provisions and Ethical Principles

To guide clinicians and policymakers, consider creating a table that synthesizes key legal provisions alongside recognized ethical principles such as autonomy, beneficence, and justice. Such a comparative overview would be a practical tool for aligning clinical practice with legal obligations.

Addressing the above suggestions would make the discussion even more relevant to a global audience and offer valuable direction to healthcare providers and policymakers.

Author Response

Thank you to the Reviewer for the valuable suggestions, please see attached the pdf file with the point by point response

Round 2

Reviewer 3 Report

Comments and Suggestions for Authors

Dear Authors,

Thank you for your thoughtful revisions. You have successfully addressed all my comments. I wish you continued success in your academic career and many valuable contributions to the field of palliative and end-of-life care.